# Immune Response after COVID-19 mRNA Vaccination in Multiple Sclerosis Patients Treated with DMTs

**DOI:** 10.3390/biomedicines10123034

**Published:** 2022-11-24

**Authors:** Valentina Mazziotti, Francesco Crescenzo, Agnese Tamanti, Caterina Dapor, Stefano Ziccardi, Maddalena Guandalini, Annalisa Colombi, Valentina Camera, Angela Peloso, Francesco Pezzini, Ermanna Turano, Damiano Marastoni, Massimiliano Calabrese

**Affiliations:** 1Department of Neurosciences, Biomedicine and Movement Sciences, University of Verona, 37134 Verona, Italy; 2Neurology Unit, “Mater Salutis” Hospital, ULSS 9 Scaligera, 37045 Legnago, Italy; 3Department of General Psychology, University of Padova, 35131 Padua, Italy; 4Department of Surgery, Dentistry, Pediatrics and Gynecology, University of Verona, 37134 Verona, Italy

**Keywords:** multiple sclerosis, disease-modifying therapies, COVID-19, vaccination, seroconversion, cellular immune response, inflammatory mediators, immunophenotype

## Abstract

The impact of disease-modifying therapies (DMTs) on the immune response to coronavirus disease-2019 (COVID-19) vaccines in persons with multiple sclerosis (pwMS) needs further elucidation. We investigated BNT162b2 mRNA COVID-19 vaccine effects concerning antibody seroconversion, inflammatory mediators’ level and immunophenotype assessment in pwMS treated with cladribine (c-pwMS, *n* = 29), fingolimod (f-pwMS, *n* = 15) and ocrelizumab (o-pwMS, *n* = 54). Anti-spike immunoglobulin (Ig)-G detection was performed by an enzyme immunoassay; molecular mediators (GrB, IFN-γ and TNF-α) were quantified using the ELLA platform, and immunophenotype was assessed by flow cytometry. ANCOVA, Student’s *t*-test and Pearson correlation analyses were applied. Only one o-pwMS showed a mild COVID-19 infection despite most o-pwMS lacking seroconversion and showing lower anti-spike IgG titers than c-pwMS and f-pwMS. No significant difference in cytokine production and lymphocyte count was observed in c-pwMS and f-pwMS. In contrast, in o-pwMS, a significant increase in GrB levels was detected after vaccination. Considering non-seroconverted o-pwMS, a significant increase in GrB serum levels and CD4+ T lymphocyte count was found after vaccination, and a negative correlation was observed between anti-spike IgG production and CD4+ T cells count. Differences in inflammatory mediators’ production after BNT162b2 vaccination in o-pwMS, specifically in those lacking anti-spike IgG, suggest a protective cellular immune response.

## 1. Introduction

The coronavirus disease 2019 (COVID-19) outbreak, caused by severe acute respiratory syndrome coronavirus 2 (SARS-CoV-2), is considered the first significant pandemic of the twenty-first century, caused global mortality of over 6 million deaths and important public health consequences. Pathogenetic mechanisms underlying COVID-19 have not been completely clarified [1], and the large-scale vaccination programs represented the most effective measures for mitigating the COVID-19 severity and diffusion worldwide [2]. COVID-19 vaccination is recommended in all persons with MS (pwMS), including those treated with disease-modifying therapies (DMTs), i.e., 70% of pwMS) [3,4,5]. DMTs are immunosuppressive or immunomodulatory drugs, and their impact on the immune response to the COVID-19 vaccine in pwMS is still controversial.

Among the vaccines currently available, the BNT162b2 mRNA-based vaccine was approved by the European Medicines Agency in December 2020 as one of those recommended for persons with MS (pwMS) [6]. The BNT162b2 vaccine induces both humoral and cell-mediated immune responses against viral spike peptides [7]. Impaired seroconversion after BNT162b2 vaccination was observed in ocrelizumab- and fingolimod-treated pwMS, whereas pwMS treated with cladribine developed a specific humoral response [8,9]. Subsequent studies partially confirmed a blunted seroconversion in ocrelizumab-treated patients but revealed that most fingolimod-treated patients developed a serological response after mRNA COVID-19 vaccination [2,10]. Furthermore, despite the absence or low presence of anti-spike immunoglobulin (Ig)-G [8], ocrelizumab-treated pwMS showed a preserved cell-mediated immune response after BNT162b2 vaccine compared to healthy people [11], characterized by increased levels of inflammatory mediators, such as granzyme B (GrB), interferon (IFN)-γ and tumor necrosis factor (TNF)-α [7]. On the contrary, other studies demonstrated a more robust cell-mediated immune activation mainly driven by T cells after the COVID-19 vaccine in pwMS treated with ocrelizumab compared with healthy controls [12].

Further research on the immune response to BNT162b2 mRNA-based vaccine in DMTs-treated pwMS might be helpful in elucidating the basis of these contradictory findings. In this longitudinal study, we evaluated the impact of DMTs on the immune response to COVID-19 vaccines in pwMS treated with cladribine (c-pwMS), fingolimod (f-pwMS) and ocrelizumab (o-pwMS) who completed the first BNT162b2 vaccination cycle. The immune response was assessed considering seroconversion vaccine-related, anti-spike immunoglobulin-G (IgG) titers, inflammatory mediators’ level and the immunophenotype profile. Specifically, the serum levels of GrB and cytokines (IFN-γ and TNF-α) were measured with a highly reproducible and ultrasensitive immune assay to identify new potential vaccine-related biomarkers and obtain a more rapid evaluation of vaccine efficacy, compatible with the routine monitoring of DMTs-treated pwMS.

## 2. Materials and Methods

### 2.1. Study Protocol Approvals and Patient Consents

The Ethics Committee of the Verona University Hospital approved the study (2413 CESC), and informed consent was obtained from all participants.

### 2.2. Patients Cohort

Ninety-eight pwMS (94 relapsing–remitting (RRMS), 4 primary-progressive (PPMS), mean age 45.0 ± 10.0 years, 64 female), followed at the MS Center of the Verona University Hospital (Italy), were enrolled from January 2021 to September 2021 and then followed over time in this 15 months longitudinal prospective study, in order to collect data relative to a possible breakthrough SARS-CoV-2 infection. Demographic, clinical and biological data of enrolled patients are reported in Table 1. Inclusion criteria for the enrollment of patients with MS were: (a) diagnosis of MS according to the most recent diagnostic criteria [13] (b) ongoing treatment with cladribine, ocrelizumab or fingolimod for at least 12 months before the study entry; (c) the willingness to undergo a two-approved dose (given 21 days apart) of the primary BNT162b2 COVID-19 vaccination cycle. For pwMS undergoing cladribine and ocrelizumab treatments, the timing of the first vaccine dose was scheduled at least three months after the last DMT administration, according to the recommendation of both the Italian and European Academy of Neurology for COVID-19 vaccination.

### 2.3. Study Protocol, Samples Collection and Analysis

The enrolled patients underwent a first blood sample at least one month prior to the first BTN162b2 vaccination and a second blood sample 20–30 days after the fulfilment of the BTN162b2 vaccination cycle in order to assess inflammatory cytokines production and immunophenotyping of blood lymphocytes pre- and post-vaccination. Concomitantly with the second blood sample collected, the SARS-CoV-2 IgG test was performed. All blood samples were collected according to the Consensus Guidelines for Blood Biobanking [14] Once collected, the blood samples were centrifuged, and the serum obtained was stored at −80 °C until use in the “MSBioB” biobank (Ethics Committee Protocol n° 66418). The health status of patients was monitored at the time of collection samples to exclude symptoms associated with infections or MS relapse.

#### 2.3.1. Anti-Spike IgG Titers

The level of SARS-CoV-2 IgG antibodies against the spike-protein was measured in all the 98 pwMS in a centralized laboratory with an automated serologic enzyme immunoassay (EIA) test (Abbott, Chicago, IL, USA). Anti-spike IgG values were expressed as Arbitrary Units (AU)/mL, and values ≥ 1.1 were considered positive, indicating seroconversion by the patient.

#### 2.3.2. Inflammatory Mediators’ Level

The cellular response was evaluated considering the expression of several inflammatory mediators, such as GrB, IFN-γ and TNF-α, in a subset of treated pwMS (*n* = 43, 15 c-pwMS, 11 f-pwMS, 17 o-pwMS). Inflammatory mediators’ levels were measured by using the ultrasensitive Ella-Simple Plex technology (ELLA microfluidic analyzer, Protein Simple; Bio-techne, San José, CA, USA), an automated enzyme-linked immunosorbent assay-ELISA. Differences between cytokines’ serum levels before and after vaccination were specifically considered in o-pwMS, dividing them into two groups: non-seroconverted and seroconverted patients.

#### 2.3.3. Immunophenotyping of Blood Lymphocytes

No sufficient lymphocytes’ immunophenotype data were available for f-pwMS; therefore the absolute counts of lymphocytes’ subsets (CD19+ B cells, CD3+, CD4+ and CD8+ T cells) were investigated only in c-pwMS and o-pwMS (total: *n* = 63). Lymphocyte subsets were assessed by flow cytometry analysis (Navios, version 1.3; Beckman Coulter, Brea, CA, USA).

### 2.4. Statistical Analysis

Analysis of covariance (ANCOVA) was performed on identified variables associated with the COVID-19 vaccine-related seroconversion. ANCOVA, with age as a covariate, followed by post-hoc comparison Tukey’s test was used to evaluate differences in anti-spike IgG seroconversion and titers, inflammatory mediators’ levels and lymphocytes’ counts among pwMS treated with different DMTs. Student’s paired *t*-test was used to assess inflammatory mediators’ levels, and lymphocytes’ count in pairwise samples of each DMTs group before and after vaccination. Pearson correlation analyses were applied to evaluate the association between anti-spike IgG titers and lymphocyte count. JASP package (6 June 2022, https://jasp-stats.org, version 0.14) was used to perform the analyses.

## 3. Results

### 3.1. Anti-spike IgG Seroconversion

The age effect on seroconversion was significantly (*p* < 0.001; Table 1). O-pwMS show a rate of seroconversion significantly lower (*n* = 20/54, 37%, *p* < 0.001) compared to c-pwMS (*n* = 25/29, 86%) and f-pwMS (*n* = 12/15, 80%; Table 1). No differences were found between c-pwMS and f-pwMS (*p* < 0.116).

### 3.2. Anti-spike IgG Titers

Among the pwMS who showed anti-spike seroconversion (*n* = 57/98, 58%), o-pwMS demonstrated on average a significant lower level of anti-spike IgG titers compared to c-pwMS (*p <* 0.001) and f-pwMS (*p =* 0.003; Table 2; Figure 1).

### 3.3. Inflammatory Mediators Levels Pre- and Post-Vaccination

No significant difference in inflammatory mediators’ production was observed between pre- and post-vaccination in c-pwMS and f-pwMS (see Appendix A), while in o-pwMS only a significant increase in GrB serum levels was found (*p* = 0.021; Figure 2A).

Considering non-seroconverted o-pwMS (*n* = 34), GrB serum levels were significantly increased after vaccination (*p =* 0.008; Figure 3A) while no difference was observed in seroconverted o-pwMS. No differences were found in IFN-γ (Figure 3B) and TNF-α (Figure 3C) although a trend towards increasing was evident) serum levels pre- and post-vaccination (see Appendix A).

### 3.4. Lymphocytes Immunophenotype Pre- and Post-Vaccination and Their Association with Anti-Spike IgG Production

Comparing lymphocytes’ immunophenotype in o-pwMS and c-pwMS, CD19+ B and CD3+ T cell subsets count did not change after vaccination in both DMT groups (see Appendix A). However, o-pwMS showed a trend of CD4+ T cells to increase after vaccination. Specifically, considering non-seroconverted o-pwMS, a significant increase in CD4+ cells count was found after vaccination, which was not observed in seroconverted o-pwMS, although even in these, an increase in CD4+ T cells was present (Figure 4, *p =* 0.040).

Therefore, only in non-seroconverted o-pwMS, a significant negative correlation was found between CD4+ T cell count post-vaccination and anti-spike IgG production below the reference cut-off (*r =* −0.438, *p =* 0.014; Figure 5).

## 4. Discussion

The ongoing COVID-19 pandemic represents one of the deadliest in history [15]. The COVID-19 mRNA-based vaccines continue to be effective against severe disease, activating a coordinated induction of both humoral- and cell-mediated immune responses [7]. Specifically, the Pfizer-BioNTech COVID-19 (BNT162b2) vaccine, containing mRNA vector encoding the prefusion spike-glycoprotein of SARS-CoV-2 [16], has been demonstrated safe in pwMS and no increased risk of MS relapse activity has been observed following vaccination [17]. However, investigating vaccine efficacy is extremely important in pwMS treated with DMTs that could make vaccination ineffective, exposing them to a higher risk of infection and severe symptoms.

In the present study, we evaluated the BNT162b2-mRNA vaccine response in terms of antibody production, inflammatory mediators’ level and lymphocytes’ immunophenotype assessment in DMTs-treated pwMS under three different MS treatments: cladribine, fingolimod and ocrelizumab.

In line with previous results, we found a fully detectable humoral response in c-pwMS [2,8] and in most f-pwMS [10,18].

Although several works demonstrate a substantially attenuated humoral response to the COVID-19 vaccination in f-pwMS [4,8,19,20], we found in fact a high seroconversion rate in f-pwMS. This result, although it is surprisingly discrepant with previous reports, is in line with other recent works, which show an humoral response in f-pwMS [10,18]. In addition to the conflicting results, interpreting this result also becomes difficult considering the small cohort of f-pwMS patients considered. For these reasons, the impact of fingolimod treatment on humoral response deserves further elucidation considering a larger cohort size.

As expected of B-cell depleting therapy [21,22], we found that most o-pwMS did not develop the antibody response to mRNA-based vaccines [2,10], although we found that the seroconversion rate of o-pwMS is overall quite high compared to other works [8,9]. However, it is worth noting that this result is in line with a recent study, which shows a similar seroconversion rate [23]. In addition, o-pwMS who developed antibodies had significantly lower antibody titers compared to c-pwMS and f-pwMS, as shown in previous works [8,18], although several subjects showed high antibody titers.

However, despite ocrelizumab treatment being the most strongly associated with a seronegative response following vaccination among all DMTs [4], as we observed, we found only one case of COVID-19 infection in the o-pwMS group, suggesting a limited increased risk of COVID-19 infection compared to the other DMTs examined (cladribine and fingolimod), in line with previous observations [2,24,25].

Furthermore, treatment duration and disease duration were not associated with spike-specific antibody production for each DMT considered.

Identification of molecular mediators’ induction following COVID-19 vaccination is therefore important to evaluate vaccine response in pwMS treated with different DMTs, especially in those who do not develop an antibody response [26]. Furthermore, the use of a straightforward and reproducible assay for molecular detection could represent a new tool for routine monitoring of vaccinated pwMS.

For all these reasons, in addition to the humoral immune response, we also explored through the ELLA platform the cellular immune response to the BNT162b2-mRNA vaccine considering the serum levels of specific inflammatory mediators, mainly released by spike-specific activated T cells, such as GrB, IFN-γ and TNF-α [27,28]. These molecular mediators have a pivotal role in the development and regulation of cellular and humoral immunity to vaccination, including the BNT162b2-mRNA vaccine [7,12,27,28]. Specifically, in our study, we found that GrB serum levels in o-pwMS (but not in the other DMTs groups) were increased after vaccination and such an increase was also more relevant in those patients who failed to generate anti-spike IgG. In the same patients, furthermore, an increasing trend in TNF-α serum levels was observed.

These results, in line with previous works [9], suggest that in o-pwMS, who underwent the B-cell depleting therapy, a cell-mediated response, presumably associated with the preservation of CD4+ and CD8+ T cells, has occurred.

In order to confirm this hypothesis, we also assessed the immunophenotypic lymphocyte profile in c-pwMS and o-pwMS, considering CD3+ T-cell subsets. We found that non-seroconverted o-pwMS showed a significant increase in CD4+ T-cell count post-vaccination compared to pre-vaccination.

Anti-spike IgG production and lymphocytes’ immunophenotype were correlated both in c-pwMS and in o-pwMS. We found a strong negative correlation between anti-spike IgG production and CD4+ cell count after vaccinations only in non-seroconverted o-pwMS, supporting previous findings and confirming a CD4+ T-cell-mediated response.

This study is not free from limitations. First of all, considering a control group would have strengthened our results, but it was not possible to enroll treatment-naïve pwMS for this work. Furthermore, the low number of patients recruited, especially f-pwMS, and of important data, in particular the lymphocytes’ immunophenotype assessment of f-pwMS, represent other limitations in our study. However, the extension of the data to a larger cohort of patients will confirm and clarify the different immunological setup in non-seroconverted pwMS.

Moreover, the study lacks the staining of T-cell activation markers such as CD25, CD69, CD71, CD95 and HLA-DR and an in vitro functional study, demonstrating the direct production of inflammatory mediators by T lymphocytes including an ELISpot assay or flow cytometry (intracellular cytokine staining analysis). Nevertheless, although the ELLA simple-plex assay used does not represent the gold standard for a functional analysis of the expression of T-cell-related molecular mediators, it is highly sensitive and reproducible, making our results robust and indicative of the inflammatory levels of these patients after vaccination.

## 5. Conclusions

Overall, this study confirms differences in anti-spike IgG production among different DMTs and provides evidence of cell-mediated immunity preservation after BNT162b2 vaccination in o-pwMS, specifically in those patients lacking anti-spike IgG antibodies. Such a hypothesis seems confirmed by the observation of a low rate (and the mild evolution) of COVID-19 infection in o-pwMS.

Finally, the evaluation of specific molecular mediators, through innovative and immediate technology, could represent a helpful tool for the routine monitoring of vaccination response in vaccinated DMTs-treated pwMS.

## Figures and Tables

**Figure 1 biomedicines-10-03034-f001:**
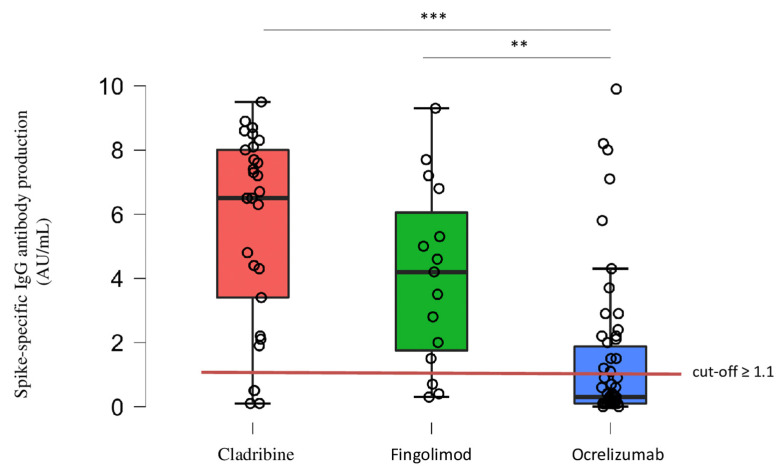
Anti-spike IgG titers. Anti-spike IgG titers (indicated as circles), obtained by EIA test (cut-off ≥ 1.1 AU/mL), was significantly lower in seroconverted o-pwMS (*n* = 20/54, 37%) compared to c-pwMS (*n* = 25/29, 86%; *** *p <* 0.001) and f-pwMS (*n* = 12/15, 80%; ** *p =* 0.003, ANCOVA followed by post-hoc pairwise comparison using the Tukey test). ** *p* < 0.01, *** *p* < 0.001.

**Figure 2 biomedicines-10-03034-f002:**
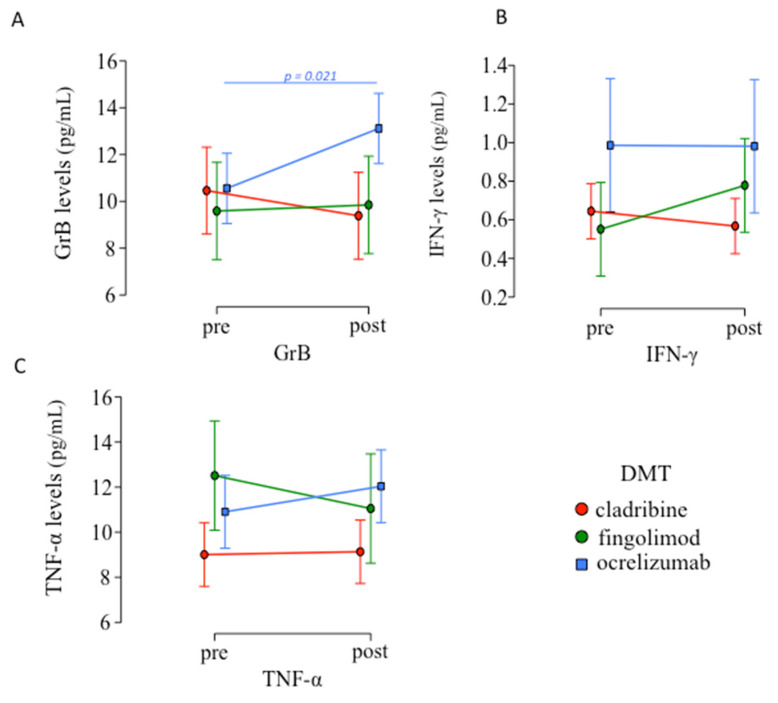
Inflammatory mediators’ levels pre- and post-vaccination. (**A**) A significant increase in GrB serum levels was observed only in o-pwMS after vaccination (mean ± SD: 13.115 ± 4.407) compared to before (10.555 ± 3.264; *p =* 0.021, Student’s paired *t*-test), while no differences were found of (**B**) IFN-γ and (**C**) TNF-α serum levels pre- and post-vaccination in each DMT.

**Figure 3 biomedicines-10-03034-f003:**
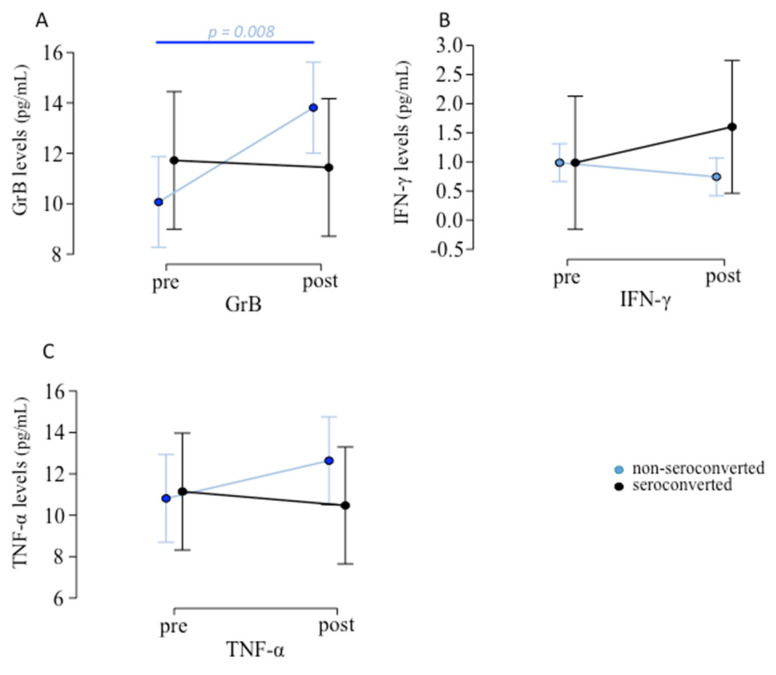
Inflammatory mediators’ levels pre- and post-vaccination in o-pwMS. (**A**) Among the 43 patients tested with ELLA, o-pwMS non-seroconverted (*n* = 13) showed GrB serum levels significantly increase after vaccination (mean ± SD: 13.813 ± 4.645 pg/mL) compared to before vaccination (10.069 ± 2.914 pg/mL; *p =* 0.008). No differences were found in (**B**) IFN-γ and (**C**) TNF-α (although a mild trend towards increasing was evident, *p =* 0.196) serum levels pre- and post-vaccination (Student’s paired *t*-test).

**Figure 4 biomedicines-10-03034-f004:**
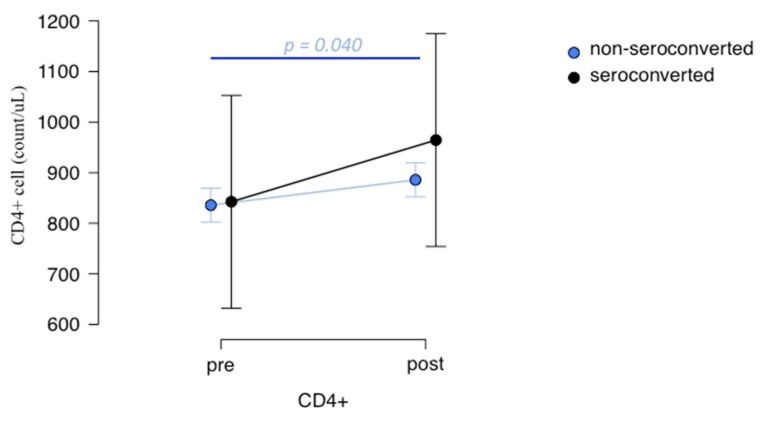
CD4+ T-cells count pre- and post-vaccination in o-pwMS. Non-seroconverted o-pwMS blood analyzed through flow cytometry (*n* = 34), showed a significant increase in CD4+ cells after vaccination (mean ± SD: 885.793 ± 352.271 cells count/uL compared to before (mean ± SD: 835.759 ± 321.184 cells count/ul, *p =* 0.040; Student’s *t*-test). No differences were found between CD4+ T-cells count pre- (842.462 ± 540.486 cells count/uL) and post-vaccination (964.462 ± 633.230 cells count/uL) in seroconverted o-pwMS (*n* = 13).

**Figure 5 biomedicines-10-03034-f005:**
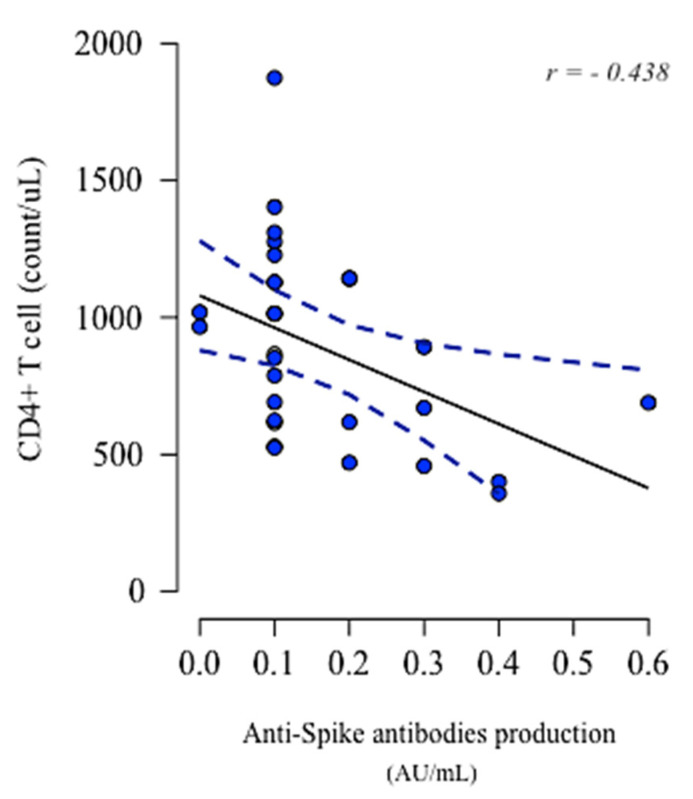
CD4+ T cells and their association with anti-spike IgG production. Negative correlation was found between CD4+ T cells (count/uL) and anti-spike IgG production (AU/mL) in non-seroconverted o-pwMS (*r =* −0.438, *p =* 0.014, Pearson correlation). Data point (in blue) indicated CD4+ T cells in individual samples and the dashed trend lines (in blue) indicate a negative correlation slope.

**Table 1 biomedicines-10-03034-t001:** Demographic, clinical and seroconversion COVID-19 vaccine-related data.

	Cladribine (*n* = 29)	Fingolimod (*n* = 15)	Ocrelizumab (*n* = 54)	*p*
Gender (F:M)	20:9	12:3	22:32	*p* = 0.107 ^a^
Age, years (mean ± SD)	41 ± 12	40 ± 10	51 ± 10	*p* = 0.023 ^a^
EDSS score (median (range))	1.5 (0–7.5)	1.0 (0–4)	6.0 (0–7.5)	*p* = 0.525 ^a^
DMTs (*n*)	29	15	54	*p* < 0.001 ^a^
Disease duration, years (mean ± SD)	7.1 ± 7.0	10.5 ± 6.4	10.7 ± 6.3	*p* = 0.571 ^a^
Treatment duration, years (mean ± SD)	1.4 ± 0.6	2.6 ± 2.0	2.2 ± 0.1	*p* = 0.739 ^a^
Seroconversion vaccine-related (*n*; %)	25; 86%	12; 80%	20; 37%	*p* < 0.001 ^b^
COVID-19 infection post-vaccine (*n*; %)	0; 0%	0; 0%	1; 1.85% ^c^	

DMTs, disease-modifying therapies; EDSS, expanded disability status scale. Anti-spike IgG production was measured by EIA test (cut-off ≥ 1.1 AU/mL) from the blood sample of pwMS treated with ocrelizumab (*n* = 54), cladribine (*n* = 29) and fingolimod (*n* = 15). ^a^ Statistical analysis was performed using ANCOVA. ^b^ Differences among the DMTs groups on seroconversion: reported here was the ANCOVA, with age as a covariate, result. ^c^ O-pwMS, who developed COVID-19 following the vaccine, was non-seroconverted.

**Table 2 biomedicines-10-03034-t002:** Anti-spike IgG titers in seroconverted pwMS.

DMTs	Seroconversion (*n*; %)	Spike-Specific IgG Antibody Titers (Mean ± SD)	Range (Min–Max)
Cladribine	25/29 (86%)	6.517 ± 2.298 AU/mL	1.900–9.500
Fingolimod	12/15 (80%)	4.992 ± 2.395 AU/mL	1.500–9.300
Ocrelizumab	20/54 (37%)	3.040 ± 2.56 AU/mL	1.100–9.900

Anti-spike IgG titers, obtained by EIA test (cut-off ≥ 1.1 AU/mL), in seroconverted pwMS.

## Data Availability

The data supporting the findings of this study are available from the corrisponding author on reasonable request.

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
