# Peer review of "Immune Response after COVID-19 mRNA Vaccination in Multiple Sclerosis Patients Treated with DMTs"

_biomedicines, 2022, doi:10.3390/biomedicines10123034_

Round 1
Reviewer 1 Report
The authors address a current topic that continues to be of great relevance to the neuroimmunological community - even though it must be said that many papers already exist on the topic of COVID-19 vaccination response under MS therapy. The paper is stringently written and the study design is comprehensible. However, the following points should be adjusted before a possible publication:
1) It would be of great interest if a more detailed panel could be added to the flow cytometric studies. Staining of T-cell activation markers such as CD25, CD69, CD71, CD95 and HLA-DR would be particularly appreciated. Addition of these measurements would add value to the work and its power.
2) It would be desirable if the authors could put their results more in the context of the already published literature.
- Compared to other work, the seroconversion rate of ocrelizumab patients is overall quite high at 37% and individual patients show surprisingly high AK titers (Fig. 1). How can this be explained? What characterized the ocrelizumab patients with high AK titers?
- Individual papers showed low seroconversion rates for fingolimod patients as well (see for example: Tallantyre EC, et al. COVID-19 Vaccine Response in People with Multiple Sclerosis. Ann Neurol. 2022;91(1):89-100.). Do the authors find an explanation for this? What is different about their current cohort?
Reviewer 2 Report
In this paper by Mazziotti et al., the authors aim at investigating the impact of disease-modifying therapies (DMTs) on the immune response to BNT162b2 mRNA COVID-19 vaccine in persons with multiple sclerosis (pwMS) treated with different pharmacological regimens such as cladribine, fingolimod and ocrelizumab.
The authors observed that GrB serum levels in o-pwMS were increased after vaccination and such increase was also more relevant in patients who failed to generate anti-spike IgG. In addition, they reported that non-seroconverted o-pwMS showed a significant increase in CD4+ T cell count post-vaccination compared to pre-vaccination. They conclude saying that in o-pwMS patients, lacking anti-spike IgG antibodies, there is a cell-mediated immunity preservation after BNT162b2 vaccination.
Although the comprehension of the impact of immunomodulatory drugs on response to vaccination is a topic of massive scientific relevance, the work presents many weaknesses.
The authors state that: "among the pwMS who showed anti-spike seroconversion, o-pwMS demonstrated a significant lower level of anti-spike IgG"... Does this mean that in the analysis showed in figure 1 have been included only the subjects in which seroconversion occurred? In the graph reported in Figure 1 seems that all patients have been included. Indeed, also in the table 2 the authors show that the lowest value of IgG (range min-max) is 0. How is this possible? If there was been a seroconversion, is it possible to have an anti-spike IgG level equal to 0?
The authors observed a significant increase in GrB serum levels in o-pwMS after vaccination when compared to the other treatments analyzed. Since, GrB secretion could be associated with activation of CD8+ cells, the authors should also evaluate the peripheral levels of this cellular subset. However, the evaluation of circulating inflammatory mediators is not sufficient to state that a cell-mediated response has occurred after vaccination. Data on peripheral levels of inflammatory mediators should be confirmed and correlated with data in vitro showing a higher activation and cytokine production of CD4+ and CD8+ T cells in response to antigen-specific stimulation.
Data showing an increase in the number of CD4+ T cells after vaccination cannot be associated with a higher activation of this cellular subset. This should be confirmed by evaluating the expression levels of activation markers on cell surface.
In addition, the authors reported that only in non-seroconverted o-pwMS, a significant negative correlation was found between CD4+ T cell count post-vaccination and anti-spike IgG production. If the authors define these subjects non-seroconverted, how is it possible to perform a correlation between CD4+ T cells and the levels of IgG?
The main problem regards the absence of a control group of non-treated pwMS without which it is difficult to establish that the phenomena observed are specifically due to the pharmacological treatment used.
